# PLANA3R: Zero-shot Metric Planar 3D Reconstruction via Feed-Forward Planar Splatting

**Changkun Liu**[1,2*]   **Bin Tan**[2*]   **Zeran Ke**[2,3]   **Shangzhan Zhang**[2,4]   **Jiachen Liu**[5]   **Ming Qian**[2,3]
**Nan Xue**[2†]   **Yujun Shen**[2]   **Tristan Braud**[1]

[1]The Hong Kong University of Science and Technology    [2]Ant Group
[3]Wuhan University    [4]Zhejiang University    [5]The Pennsylvania State University

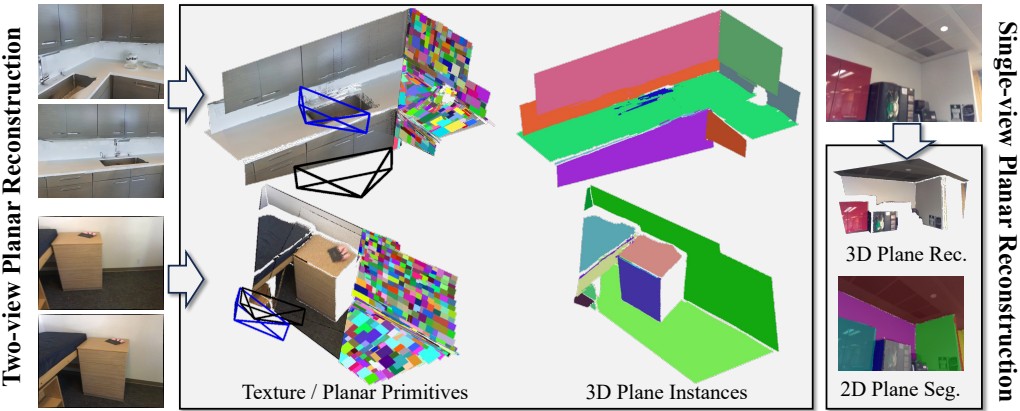

Figure 1: Our proposed PLANA3R learns to predict planar 3D primitives and metric-scale relative poses, providing a compact 3D representation of two-view input images with accurate pose estimation, surface geometry, and semantically meaningful 2D & 3D planar segmentation—all in one pass. As a side output, our model also performs well on single-view planar 3D reconstruction.

## Abstract

This paper addresses metric 3D reconstruction of indoor scenes by exploiting their inherent geometric regularities with compact representations. Using planar 3D primitives – a well-suited representation for man-made environments – we introduce PLANA3R, a pose-free framework for metric Planar 3D Reconstruction from unposed two-view images. Our approach employs Vision Transformers to extract a set of sparse planar primitives, estimate relative camera poses, and supervise geometry learning via planar splatting, where gradients are propagated through high-resolution rendered depth and normal maps of primitives. Unlike prior feedforward methods that require 3D plane annotations during training, PLANA3R learns planar 3D structures without explicit plane supervision, enabling scalable training on large-scale stereo datasets using only depth and normal annotations. We validate PLANA3R on multiple indoor-scene datasets with metric supervision and demonstrate strong generalization to out-of-domain indoor environments across diverse tasks under metric evaluation protocols, including 3D surface reconstruction, depth estimation, and relative pose estimation. Furthermore, by formulating with planar 3D representation, our method emerges with the ability for accurate plane segmentation. The project page is available at: `https://lck666666.github.io/plana3r/`.

---

*Equal Contribution.

†Corresponding author.

39th Conference on Neural Information Processing Systems (NeurIPS 2025).

# 1 Introduction

Indoor environments are the primary setting where humans spend most of their daily lives. Yet, computationally creating digital twins of these 3D spaces from captured images remains challenging. Factors such as the difficulty of accurate camera pose estimation from indoor images [28, 11, 1] and structural distortions in the resulting 3D reconstructions [22, 12, 21] hinder the development of robust, accurate, and user-friendly solutions for replicating indoor scenes in the digital world.

As indoor scenes are typically rich in planar structures such as floors, ceilings, and walls, as well as planar furniture like tables and cabinets, planar primitives are well-suited representations for the accurate 3D reconstruction of indoor scenes. As a result, there has been significant interest among the research community in planar 3D reconstruction in recent years. Planar reconstruction approaches include feedforward solutions in monocular [40, 16, 27, 24, 18, 42] and two-view [11, 1, 28] settings, and per-scene optimization approaches [29, 38, 3, 9] that leverage posed multi-view inputs with the assistance of the feedforward models were studied. However, these approaches face two key limitations:

- **Annotation dependence for feedforward methods**: Learning feedforward models [36, 24, 28] typically requires accurate plane masks and 3D plane annotations from monocular or binocular inputs. This reliance on dense annotations limits generalizability and thus impairs zero-shot performance on out-of-distribution data.

- **Pose dependence for optimization-based methods**: Per-scene optimization techniques [29, 9, 38, 3] depend on accurately posed multi-view, densely-captured images, which are not always available and would lead to either undermining reconstruction quality or limited usage scenarios. Furthermore, these methods cannot handle several sparse pose-free views or just a pair of pose-free views.

This paper focuses on two-view planar 3D reconstruction, one of the most fundamental tasks in structured indoor scene modeling. We aim to eliminate the reliance on dense plane-level annotations and accurate multi-view camera poses, addressing the limitations discussed above. Inspired by recent advancements in 3D foundation models [33, 14, 31, 32, 30, 37, 41], we show that feedforward, pose-free, and zero-shot generalizable planar 3D reconstruction from unposed stereo pairs is both feasible and effective through our proposed method, PLANA3R.

PLANA3R is built upon Vision Transformers [7], which process stereo image pairs to jointly learn 3D planar primitives and relative camera poses at scale. Rather than relying on explicit planar 3D annotations, we leverage depth and normal maps[3], which are more readily available in existing two-view datasets at scale, as supervision signals to train the Transformer-based model. To this end, we adopt the differentiable planar rendering technique from PlanarSplatting [29] to generate high-resolution rendered depth and normal maps, and then compare them with the corresponding ground truth to guide learning through gradient-based optimization. Once the model is trained, our method generates a set of 3D planar primitives that approximate indoor scenes far more efficiently than per-scene optimization methods [29, 9, 38, 3]. As shown in Fig. 1, our trained PLANA3R computes metric pose and planar primitives in a single feed-forward inference, resulting in a compact representation of indoor scenes with accurate geometry and meaningful semantics.

Given that indoor environments are constructed by humans, and the dimensions of human-scale objects tend to follow similar distributions across different scenes, we consider that planar 3D representation inherently possesses the potential to predict metric 3D geometry directly. Therefore, during training data preparation, we pay particular attention to preserving the metric scale of both depth maps and relative camera poses. In total, four million training data from ScanNetV2 [4], ScanNet++ [39], ARKitScenes [5], and Habitat [23] have been prepared to train our PLANA3R model with metric scales.

PLANA3R exhibits a strong generalization ability to out-of-domain indoor environments in terms of depth estimation, 3D surface reconstruction, and relative pose estimation. Benefiting from the planar-based representation we used, our model also empowers the capacity to provide promising instance-wise planar segmentation, enriching the semantics of 3D reconstruction without the need for

---

[3]For the dataset that only contains depth maps, the Metric3Dv2 [10] is applied to generate pseudo labels as the normal maps. Please refer to Sec. 4.2 for details.

plane masks. Furthermore, we argue that indoor scenes are particularly well-suited for training metric 3D vision foundation models. The regular geometry and semantic consistency of indoor environments provide an ideal context for developing models that generalize across scenes and accurately estimate metric information.

## 2 Related Work

In this section, we briefly summarize the recent developments on the 3D reconstruction of indoor scenes in two aspects of planar representation and feedforward modeling.

### 2.1 Indoor Planar 3D Reconstruction

Given the strong planarity of indoor scenes, planar surface reconstruction has been extensively studied in the literature [17, 27, 16, 24, 1, 40, 11, 28, 34, 36, 29, 9, 38]. Among existing methods, feedforward approaches [24, 27, 36, 28, 11] are particularly appealing due to their simplified computation pipelines powered by neural networks. However, most of these methods focus on in-domain testing using small or moderate-scale datasets. A major limitation is the scarcity of large-scale, high-quality 3D planar annotations, which constrains supervision and limits the generalization ability of models trained on such data.

For instance, methods like SparsePlanes [11], PlaneFormers [1], and NOPE-SAC [28] rely on single-view 3D plane annotations and two-view 3D plane correspondences as supervision, restricting usable data sources to ScanNetV2 [4] and Matterport3D [2] datasets and requiring complex data preprocessing pipelines. To address this, our work improves data accessibility by leveraging PlanarSplatting [29] as an alternative to the manually-annotated planar supervisions. This makes it possible to train models on large-scale 3D datasets by learning sparse planar primitives without requiring explicit plane-level annotations.

### 2.2 Feed-forward Stereo Foundation Models for 3D Reconstruction

The rise of large-scale training has significantly transformed the landscape of 3D reconstruction [33, 14, 32, 41, 31, 30, 37]. Recently, DUSt3R [33] introduced a feedforward framework that predicts dense point clouds from stereo image pairs without requiring known camera poses. Building on this, MASt3R [14] further enhances performance by generating point clouds in metric space. Several follow-up works have since explored ways to improve both scene geometry estimation [32, 30] and camera pose prediction [6, 41]. However, these methods typically assume that scene geometry can be effectively represented by densely sampled 3D points, which introduces redundancy when modeling structured environments. In contrast, we adopt more abstract planar primitives to express structured scenes with the best pursuit of compactness and efficiency in representation.

While our method shares architectural similarities with DUSt3R [33] and MASt3R [14], notably the use of Vision Transformers [7], we depart from their dense point-based representations. Instead, we focus on exploiting the strong planar regularities characteristic of indoor scenes by adopting a structured planar representation. We demonstrate that this formulation, based on the use of sparse planar primitives, can represent structured scenes more compactly and efficiently while also enabling the extraction of instance-level planar semantics.

## 3 Method

PLANA3R is a transformer-based model for two-view metric 3D reconstruction, using sparse 3D planar primitives as a scene representation. Given two pose-free images from the same scene, along with known intrinsics, PLANA3R predicts their relative camera pose and infers a set of 3D planar primitives in a single feed-forward pass. We build upon planar primitives introduced in PlanarSplatting [29], and leverage its CUDA-based differentiable renderer for supervision. PlanarSplatting is a core component of PLANA3R, providing ultra-fast and accurate reconstruction of planar surfaces in indoor scenes from multi-view images. Instead of detecting or matching planes in 2D or 3D, it directly splats 3D planar primitives into dense depth and normal maps via differentiable, CUDA-accelerated rasterization.

This allows PLANA3R to be trained directly from monocular depth and normal labels, without requiring explicit plane annotations. Sparse planar primitives offer a more compact and semantically meaningful alternative to dense point clouds or 3D Gaussian Splatting (3DGS) [13], particularly in structured indoor environments. They approximate scene geometry with high fidelity and support the rendering of dense, consistent depth and normal maps. Moreover, the explicit structure of primitives facilitates downstream tasks such as plane extraction and segmentation.

The predicted primitives can be further merged into coherent planar surfaces to support both 3D reconstruction and segmentation. We detail the task setup and notation in Sec. 3.1, describe our architecture in Sec. 3.2, outline training objectives in Sec. 3.3, and explain the primitive merging process in Sec. 3.4.

## 3.1 Task Setting and Notation

An overview of PLANA3R is shown in Fig. 2. The input consists of two images $I^1, I^2 \in \mathbb{R}^{3 \times H \times W}$ with camera intrinsics $\mathbf{K}^1$ and $\mathbf{K}^2$. Our goal is to train a network $\mathcal{F}$ outputs a set of sparse 3D planar primitives and the 6-DoF relative camera pose $P_{\text{rel}}$. $P_{\text{rel}}$ is represented by the quaternion $\mathbf{q} \in \mathbb{R}^4$ and translation $\mathbf{t} \in \mathbb{R}^3$:

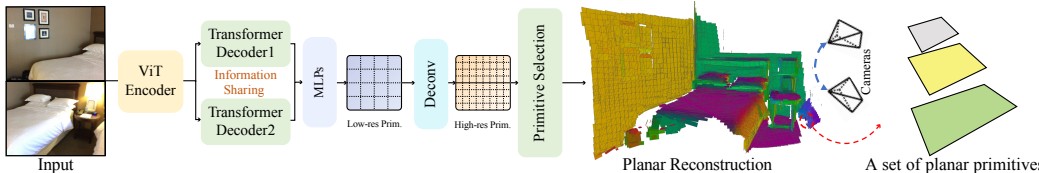

Figure 2: Overview of our PLANA3R. Given two images captured from the same scene, PLANA3R outputs a set of 3D planar primitives and 6-DoF relative camera pose $P_{\text{rel}}$ in metric scale. PLANA3R does not perform per-pixel primitive prediction. Instead, it employs a deconvolution network to predict primitives at two distinct resolutions, based on the patch divisions from the ViT encoder. During both training and inference phases, we selectively integrate low-resolution and high-resolution primitives, resulting in a highly compact and expressive representation. Note that even for high-resolution primitives, their total number is only $\frac{H}{8} \times \frac{W}{8}$, which remains highly sparse.

$$\mathcal{F} : \left\{ (I^i, \mathbf{K}^i) \right\}_{i=1,2} \Rightarrow \left\{ (d^i_\pi, \mathbf{r}^i_\pi, \mathbf{q}^{i,j}_\pi) \right\}^{N_i}_{i=1,2; \ j=1}, \ P_{\text{rel}}, \tag{1}$$

where $\mathbf{q}^{i,j}_\pi$ denotes the quaternion of a planar primitive generated from image $I^i$, represented in the coordinate frame of the camera $j$. $d^i_\pi \in \mathbb{R}^1_+$ denotes the depth of a planar primitive center associated with image $I^i$. $N_i$ denotes the number of planar primitives associated with image $I^i$. $\mathbf{r}_\pi = (\mathbf{r}^x_\pi, \mathbf{r}^y_\pi), \mathbf{r}_\pi \in \mathbb{R}^2_+$ denotes the plane radii. $\mathbf{r}^i_\pi$ refers to the radii array of a planar primitive associated with image $I^i$. In our training setup, we fix $j = 1$, treating $I^1$ as the reference frame. Accordingly, the ground-truth camera pose of $I^1$ is set to the $4 \times 4$ identity matrix. Then, the primitive centers $\mathbf{c}^i_\pi \in \mathbb{R}^3$ from both $I^1$ and $I^2$ are transformed into the coordinate frame of the camera $j = 1$ using the corresponding intrinsics $\mathbf{K}^i$, depth $d^i_\pi$, and camera poses. Following PlanarSplatting [29], we define the same positive direction of the X-axis/Y-axis of the 3D planar primitive and calculate the normal of the planar primitive $\mathbf{n}_\pi \in \mathbb{R}^3$ as:

$$\mathbf{n}_\pi = \mathbf{R}(\mathbf{q}_\pi)[0, 0, 1]^T, \tag{2}$$

where $\mathbf{R}(\mathbf{q}_\pi)$ converts $\mathbf{q}_\pi$ to the rotation matrix.

Input images $\{I^i\}_{i=1,2}$ are first encoded in a Siamese fashion using a ViT encoder [7], producing feature maps $\{F^i\}_{i=1,2} \in \mathbb{R}^{\frac{H}{16} \times \frac{W}{16} \times D_{\text{enc}}}$. These features are then processed by two transformer decoders with cross-attention to produce low-resolution decoder embeddings $\{G^i_{\text{low}}\}_{i=1,2} \in \mathbb{R}^{\frac{H}{16} \times \frac{W}{16} \times D_{\text{dec}}}$. A separate head regresses the relative camera pose $P_{\text{rel}}$ from the concatenated low-resolution features $\{G^i_{\text{low}}\}_{i=1,2}$. This partial architecture of our network $\mathcal{F}$ is inspired by some stereo 3D vision models [33, 14, 6, 35]. The core innovation of our method lies in the sparse primitive prediction architecture outlined in Sec. 3.2 and Sec. 3.3.

Compared to per-pixel 3D Gaussian Splatting and dense 3D point regression methods, our approach leverages the inherent compactness of planar primitives. Many pixels naturally form coherent planar regions in structured indoor environments, making per-pixel representations unnecessarily redundant. Although non-planar or texture-rich areas may require finer approximations with smaller primitives, such cases are relatively rare. To achieve a more compact and efficient geometric representation using fewer primitives, we propose a *hierarchical primitive prediction architecture* (HPPA) to fit the scene using planar primitives, enabling compact modeling of scene geometry with sparse primitives.

## 3.2 Hierarchical Primitive Prediction Architecture

From $\{G_{\text{low}}^i\}_{i=1,2}$, three regression heads first predict low-resolution patched planar primitives $\left\{ (d_\pi^i,\ \mathbf{r}_\pi^i,\ \mathbf{q}_\pi^{i,j}) \right\}_{i=1,2;\ j=1}^{\frac{H}{16} \times \frac{W}{16}}$. To obtain higher-resolution features, we apply a deconvolution network to upsample $\{G_{\text{low}}^i\}_{i=1,2}$, generating $\{G_{\text{high}}^i\}_{i=1,2} \in \mathbb{R}^{\frac{H}{8} \times \frac{W}{8} \times D_{\text{dec}}}$. The same three heads then regress high-resolution patched planar primitives $\left\{ (d_\pi^i,\ \mathbf{r}_\pi^i,\ \mathbf{q}_\pi^{i,j}) \right\}_{i=1,2;\ j=1}^{\frac{H}{8} \times \frac{W}{8}}$. The primary challenge then becomes determining which image regions require low-resolution planar primitives and which benefit from higher-resolution primitives. To address this, we propose a simple yet effective heuristic that avoids the need for additional learning.

Our approach is based on the observation that regions exhibiting significant variations in normals require a greater number of smaller planar primitives for accurate fitting, whereas areas with minimal changes in normals can be effectively represented by fewer planar primitives with larger radii. For each input image, using $d_\pi$ in predicted primitives, we derive the patched depth maps: $\mathbf{D}_{\text{low}}^{\text{patch}} \in \mathbb{R}^{1 \times \frac{H}{16} \times \frac{W}{16}}$ and $\mathbf{D}_{\text{high}}^{\text{patch}} \in \mathbb{R}^{1 \times \frac{H}{8} \times \frac{W}{8}}$. Similarly, using $q_\pi$ in predicted primitives and Eq. (2), two different patched normal maps are obtained: $\mathbf{N}_{\text{low}}^{\text{patch}} \in \mathbb{R}^{3 \times \frac{H}{16} \times \frac{W}{16}}$ and $\mathbf{N}_{\text{high}}^{\text{patch}} \in \mathbb{R}^{3 \times \frac{H}{8} \times \frac{W}{8}}$. We compute the gradient magnitude for each pixel in the low-resolution predicted normal patches $\mathbf{N}_{\text{low}}^{\text{patch}}$ of size $\frac{H}{16} \times \frac{W}{16}$, and selectively use the high-resolution planar primitives of size $\frac{H}{8} \times \frac{W}{8}$ only for those pixels whose gradients exceed a predefined threshold $g_{\text{th}}$ during rendering. To combine the low- and high-resolution primitives, we use binary masks to merge only the valid patches from both resolutions, rather than directly using all predicted primitives. We provide more details in Fig. 12 (b).

## 3.3 Training Losses and Training Strategies

For input images $\{I^i\}_{i=1,2}$, PLANA3R generates planar primitives at both low and high resolutions. Specifically, the low-resolution primitives are represented as $\left\{ (d_\pi^i, \mathbf{r}_\pi^i, \mathbf{q}_\pi^{i,j}) \right\}_{i=1,2;\ j=1}^{\frac{H}{16} \times \frac{W}{16}}$, while the high-resolution primitives are given by $\left\{ (d_\pi^i, \mathbf{r}_\pi^i, \mathbf{q}_\pi^{i,j}) \right\}_{i=1,2;\ j=1}^{\frac{H}{8} \times \frac{W}{8}}$. Due to the randomness of initialization at the beginning of training, the initial planar primitives often lie outside the viewable range defined by the ground-truth (GT) camera poses. As a result, directly applying the differentiable planar primitive rendering from PlanarSplatting [29] becomes ineffective for optimizing these learnable primitives. To address these challenges and facilitate training, we introduce a patch loss designed to stabilize primitive positioning and orientation:

$$\mathcal{L}_*^{\text{patch}} = \alpha_1 \left\| 1 - \left( \mathbf{N}_*^{\text{patch}} \right)^\top \mathbf{N}_*^{\text{r, gt}} \right\|_1 + \alpha_1 \left\| \mathbf{N}_*^{\text{patch}} - \mathbf{N}_*^{\text{r, gt}} \right\|_1 + \alpha_2 \left\| \mathbf{D}_*^{\text{patch}} - \mathbf{D}_*^{\text{r, gt}} \right\|_1, \quad (3)$$

where $* \in \{\text{low}, \text{high}\}$, $\alpha_1$ and $\alpha_2$ are loss weights. Here, $\mathbf{D}_*^{\text{r, gt}}$ and $\mathbf{N}_*^{\text{r, gt}}$ are derived by resizing the ground-truth depth maps $\mathbf{D}^{\text{gt}} \in \mathbb{R}^{1 \times H \times W}$ and normal maps $\mathbf{N}^{\text{gt}} \in \mathbb{R}^{3 \times H \times W}$ to dimensions $\frac{H}{16} \times \frac{W}{16}$ and $\frac{H}{8} \times \frac{W}{8}$, respectively. The patch loss in the initial warm-up training stage aids in stabilizing the spatial positioning and orientation of the primitives.

After the warm-up phase, we introduce a rendering loss. We render depth and normal maps from both primitive sets. Unlike the patch loss in Eq. (3), which supervises only depth and orientation, the rendering loss in Eq. (4) compares rendered and ground-truth maps using PlanarSplatting's rasterization at full resolution ($H \times W$). This enables gradient flow to refine primitive radii and improves geometric fidelity. Specifically, $\mathbf{N}_{\text{low}}^{\text{render}} \in \mathbb{R}^{3 \times H \times W}$ and $\mathbf{D}_{\text{low}}^{\text{render}} \in \mathbb{R}^{1 \times H \times W}$ are rendered from $\frac{H}{16} \times \frac{W}{16}$ low-resolution primitives, while $\mathbf{N}_{\text{high}}^{\text{render}} \in \mathbb{R}^{3 \times H \times W}$ and $\mathbf{D}_{\text{high}}^{\text{render}} \in \mathbb{R}^{1 \times H \times W}$ originate from $\frac{H}{8} \times \frac{W}{8}$ high-resolution primitives. As discussed in Sec. 3.2, during the inference stage,

rendering all regions of the image using all high-resolution primitives is redundant. Therefore, during training, we select some low-resolution primitives and some high-resolution primitives for rendering. We compute the gradient magnitude for each pixel in low-resolution $\mathbf{N}_{\text{low}}^{\text{patch}}$ and use high-resolution planar primitives only for those pixels whose gradients exceed a predefined threshold $g_{\text{th}}$ during rendering. $\mathbf{N}_{\text{selected}}^{\text{render}} \in \mathbb{R}^{3 \times H \times W}$ and $\mathbf{D}_{\text{selected}}^{\text{render}} \in \mathbb{R}^{1 \times H \times W}$ are rendered from $N$ selected primitives, $\frac{H}{16} \times \frac{W}{16} \leq N \leq \frac{H}{8} \times \frac{W}{8}$.

To optimize these learnable planar primitives, our PLANA3R is trained in a supervised manner based on the differentiable planar primitive rendering in PlanarSplatting:

$$\mathcal{L}_*^{\text{render}} = \beta_1 \left\| 1 - \left( \mathbf{N}_*^{\text{render}} \right)^\top \mathbf{N}^{\text{gt}} \right\|_1 + \beta_1 \left\| \mathbf{N}_*^{\text{render}} - \mathbf{N}^{\text{gt}} \right\|_1 + \beta_2 \left\| \mathbf{D}_*^{\text{render}} - \mathbf{D}^{\text{gt}} \right\|_1, \quad (4)$$

where $* \in \{\text{low}, \text{high}, \text{selected}\}$ and $\beta_1, \beta_2$ balance the loss magnitudes for stable training.

For the predicted relative pose $P_{\text{rel}} = [\mathbf{t}, \mathbf{q}]$, we use MSE loss and relative angle loss to provide supervision:

$$\mathcal{L}^{\text{pose}} = \gamma_1 \left\| \mathbf{t}^{\text{gt}} - \mathbf{t} \right\|_1 + \gamma_2 \left\| \mathbf{q}^{\text{gt}} - \frac{\mathbf{q}}{\|\mathbf{q}\|} \right\|_1 + \gamma_3 (1 - \frac{\mathbf{t} \cdot \mathbf{t}^{\text{gt}}}{\|\mathbf{t}\| \|\mathbf{t}^{\text{gt}}\|}), \quad (5)$$

where $\gamma_1, \gamma_2, \gamma_3$ are loss weights for different items.

Note that our model focuses on metric reconstruction. Therefore, we do not apply normalization to the training labels. Instead, we use metric depth maps and metric poses for supervision in Eq. (3), Eq. (4), and Eq. (5).

### 3.4  3D Plane Merge

Given a pair of input images, once the collection of 3D planar primitives is predicted, we perform a similar merging in PlanarSplatting by setting thresholds for the normal and distance errors between each pair of primitives. This process enables the extraction of semantic information for each plane and yields the final planar surface reconstruction. We provide more details in Appendix A.2.

## 4  Experiment

### 4.1  Implementation Details

We initialize the ViT encoder and the transformer decoder's part of PLANA3R model with DUSt3R's pre-trained 512-DPT weights. Training is performed using the AdamW optimizer [19] with a learning rate starting at $1 \times 10^{-4}$ and decaying to $1 \times 10^{-6}$. The model is trained for a total of 256 GPU-days on NVIDIA H20 GPUs, with a per-GPU batch size of 6. Training starts with a one-epoch warm-up phase that optimizes only the losses in Eq. (3) and Eq. (5), followed by 10 epochs incorporating all three losses at an input resolution of $512 \times 384$. During both training and testing, we set the gradient threshold $g_{\text{th}}$ for merging high- and low-resolution primitives to $0.5$. Additional details are provided in the *supplementary materials*.

### 4.2  Datasets

Since PLANA3R targets structured indoor scenes, we train it on a combination of four public indoor-scene datasets: ScanNetV2 [4], ScanNet++ [39], ARKitScenes [5], and Habitat [23]. From these datasets, we construct approximately four million image pairs. Pseudo GT normal maps are generated using Metric3Dv2 [10], while GT depth maps are directly taken from the datasets. For evaluation, we use ScanNetV2, Matterport3D [2], NYUv2 [20], and Replica [26] as test sets. Among these test sets, except for ScanNetV2, the remaining three datasets demonstrate the generalization capability of our model across different datasets. We also evaluate on 7Scenes [25] in Appendix A.1.

### 4.3  Baselines and Evaluation Metrics

We evaluate our PLANA3R against state-of-the-art (SOTA) planar reconstruction methods across multiple tasks, including 3D reconstruction, pose estimation, depth estimation, and plane segmentation, using diverse scene types and image pairs. These comprehensive evaluations

demonstrate our method's superior performance in both geometric accuracy (metric 3D reconstruction, depth estimation, and two-view relative pose estimation) and semantic understanding (plane segmentation). We primarily compare PLANA3R with two-view and single-image planar 3D reconstruction methods [11, 24, 28]. We also compare with the currently popular stereo metric 3D vision foundation model, MASt3R [14], which uses dense point clouds as the scene representation.

### 4.3.1 Metric Two-view Reconstruction

We evaluate the geometric quality of reconstructed 3D planes using Chamfer Distance and F-score on the ScanNetV2 and Matterport3D datasets. Given a pair of input images, PLANA3R predicts a collection of planar primitives, which are subsequently merged into the planar surface reconstruction using the approach described in Sec. 3.4.

We conduct extensive experiments across a wide range of scenes to evaluate the effectiveness of our method in two-view 3D reconstruction and relative pose estimation. For ScanNetV2, we follow the training and testing splits defined by NOPE-SAC [28], evaluating 4051 image pairs from 303 scenes. For Matterport3D, we evaluate 6083 image pairs from 13 challenging scenes. As shown in Tab. 1, PLANA3R achieves SOTA performance on ScanNetV2. Remarkably, despite never being trained on Matterport3D, PLANA3R outperforms prior planar reconstruction methods [11, 24, 28] that were specifically trained on this dataset, highlighting the strong zero-shot generalization capability of PLANA3R across diverse indoor environments.

We further evaluate relative camera pose estimation on ScanNetV2 and Matterport3D using the same image pairs evaluated in reconstruction. Pose accuracy is measured by the metric translation error (in meters) and rotation error (in degrees). As shown in Tab. 1, both MASt3R and our PLANA3R significantly outperform prior learning-based planar reconstruction methods [28, 11, 1] in terms of pose estimation accuracy.

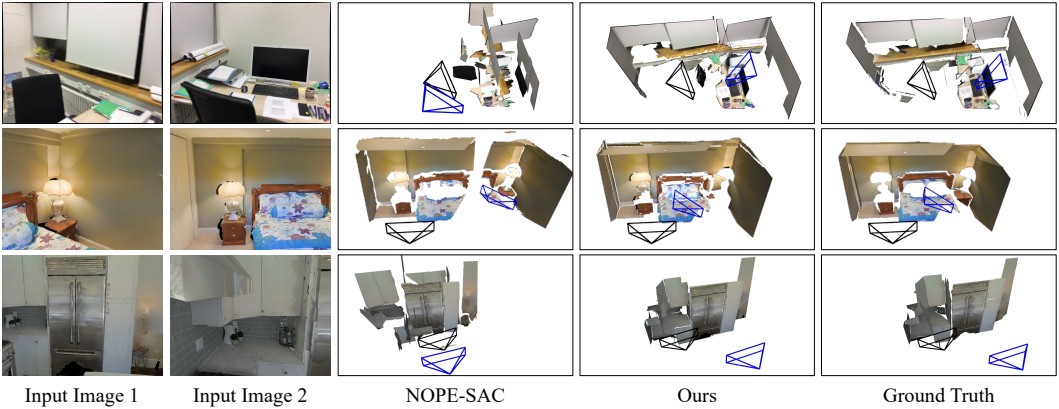

| Input Image 1 | Input Image 2 | NOPE-SAC | Ours | Ground Truth |

Figure 3: Comparisons of two-view 3D planar reconstruction on the ScanNetV2 [4] (the first row) and the Matterport3D [2] (the last two rows) datasets.

### 4.3.2 Metric Monocular Depth Estimation

Directly rendering depth from the predicted planar primitives provides an effective means of evaluating the quality of our geometric fitting. This process does not require merging the primitives and can be performed with a single feed-forward pass. We evaluate the metric monocular depth estimation using the NYUv2 [20] dataset. For our experiments, we use the image splits defined in [24, 15] with 654 test frames. We input two identical images into PLANA3R and use the CUDA-based differentiable planar primitive rendering module in PlanarSplatting to generate a depth map of the predicted primitives from one of the views.

Since our PLANA3R has never seen the scenes of NYUv2 during training, this dataset can well demonstrate the generalization ability of our model for out-of-domain images. As shown in Tab. 2, PLANA3R demonstrates strong zero-shot metric depth estimation performance, surpassing both prior learning-based planar reconstruction methods and the metric 3D vision foundation model MASt3R.

Table 1: Quantitative comparison of two-view planar reconstruction and relative camera pose estimation. The best results are in **bold**.

| Method | Translation (m) | | | | | Rotation (°) | | | | | Chamfer↓ | F-score↑ |
|---|---|---|---|---|---|---|---|---|---|---|---|---|
| | Med. | Mean | <=1 | <=0.5 | <=0.1 | Med. | Mean | <=30 | <=15 | <=5 | | |
| | | | | | | ScanNetV2 dataset | | | | | | |
| SparsePlanes [11] | 0.56 | 0.81 | 73.7 | 44.6 | – | 15.46 | 33.38 | 70.5 | 48.7 | – | – | – |
| PlaneFormers [1] | 0.55 | 0.81 | 75.3 | 45.5 | – | 14.34 | 32.08 | 73.2 | 52.1 | – | – | – |
| NOPE-SAC [28] | 0.41 | 0.65 | 82 | 59.1 | 5.01 | 8.27 | 22.12 | 82.6 | 73.2 | 25.03 | 0.26 | 61.86 |
| MASt3R [14] | 0.11 | 0.19 | 97.65 | 93.98 | 47.37 | 2.17 | 6.67 | 95.04 | 94.08 | 84.37 | 0.21 | 74.92 |
| **ours** | **0.07** | **0.13** | **98.62** | **97.16** | **67.91** | **2.01** | **3.16** | **99.23** | **98.89** | **93.14** | **0.11** | **92.52** |
| | | | | | | Matterport3D dataset | | | | | | |
| SparsePlanes [11] | 0.62 | 1.10 | 67.32 | 40.67 | 3.67 | 7.27 | 22.11 | 84.15 | 73.60 | 36.94 | 0.47 | 48.59 |
| NOPE-SAC [28] | 0.53 | 0.91 | 73.60 | 47.82 | 4.14 | 2.79 | 13.81 | 89.71 | 87.41 | 69.83 | 0.38 | 54.96 |
| MASt3R [14] | 0.41 | 0.59 | 85.20 | 58.87 | 8.55 | **0.98** | 4.66 | 96.58 | 96.09 | **92.59** | 0.49 | 30.01 |
| **ours** | **0.24** | **0.45** | **92.78** | **78.83** | **15.07** | 2.00 | **4.49** | **98.01** | **97.19** | 89.82 | **0.32** | **56.63** |

These results highlight the potential of using sparse 3D planar primitives as an efficient and effective scene representation, in contrast to methods relying on heavy DPT heads [21] and dense point clouds.

Table 2: Quantitative comparison of metric monocular depth estimation on the NYUv2 dataset. The best results are in **bold**.

| Method | PlaneNet [15] | PlaneAE [40] | PlaneRCNN [16] | PlaneTR [27] | PlaneRecTR [24] | MASt3R [14] | **Ours** |
|---|---|---|---|---|---|---|---|
| Rel ↓ | 0.239 | 0.205 | 0.183 | 0.195 | 0.157 | 0.152 | **0.132** |
| $\log_{10}$ ↓ | 0.124 | 0.097 | 0.076 | 0.095 | 0.073 | **0.058** | 0.059 |
| RMSE ↓ | 0.913 | 0.820 | 0.619 | 0.803 | 0.547 | 0.51 | **0.463** |
| $\delta_1$ ↑ | 53.0 | 61.3 | 71.8 | 63.3 | 74.2 | 83.0 | **86.4** |
| $\delta_2$ ↑ | 78.3 | 87.2 | 93.1 | 88.2 | 94.2 | 95.6 | **96.3** |
| $\delta_3$ ↑ | 90.4 | 95.8 | 98.3 | 96.1 | **99.0** | 98.7 | 98.4 |

### 4.3.3 3D Plane Segmentation

The previous experiments demonstrate the geometric accuracy of our model. Here, we show that PLANA3R can perform zero-shot plane-level semantic segmentation without plane annotations. By merging predicted planar primitives, it infers semantically meaningful 3D planes. We evaluate segmentation quality using standard metrics: Variation of Information (VOI), Rand Index (RI), and Segmentation Covering (SC).

**Single-view 3D Plane Segmentation.** Following PlaneRCNN [16], we generate 3D plane GT labels on the Replica dataset [26] by first fitting planes to the GT mesh using RANSAC [8], and then splitting co-planar regions with different semantics into separate plane instances. A total of 498 images are sampled for single-view plane-level segmentation evaluation. Similar to monocular depth estimation, we input two identical images into PLANA3R, and then we merge the output primitives. Tab. 3 and Fig. 4 show that our PLANA3R performs better than the method trained with plane annotation in both plane segmentation and 3D reconstruction. We present more visualization results in the *supplementary materials* and conduct tests on the 7-Scenes [25] dataset.

**Two-view 3D plane segmentation.** We also evaluate plane segmentation in the two-view setting. As shown in Fig. 5, our method produces results that are much better than NOPE-SAC. More results are provided in the *supplementary materials*.

Table 3: Quantitative comparison of single-view planar reconstruction on the Replica dataset [26]. The best results are in **bold**.

| Method | Plane Segmentation | | | Plane Recall (depth) | | Plane Recall (normal) | |
|---|---|---|---|---|---|---|---|
| | RI↑ | VOI↓ | SC↑ | @0.1m | @0.6m | @5° | @30° |
| PlaneRecTR[24] | 0.85 | 1.81 | 0.58 | 1.78 | 13.46 | 3.79 | 18.92 |
| **Ours** | **0.89** | **1.62** | **0.63** | **7.79** | **30.74** | **28.52** | **36.31** |

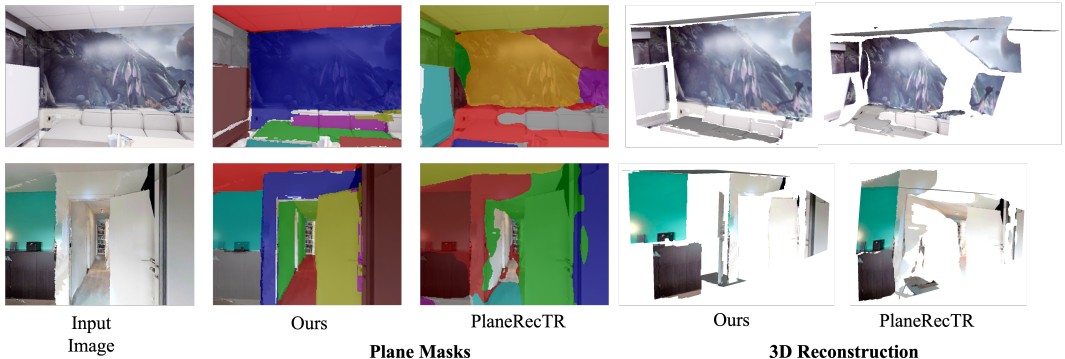

| Input
Image | Ours | PlaneRecTR | Ours | PlaneRecTR |
| --- | --- | --- | --- | --- |
| | **Plane Masks** | | **3D Reconstruction** | |

Figure 4: Comparisons of single-view plane segmentation and 3D reconstruction on the Replica [26].

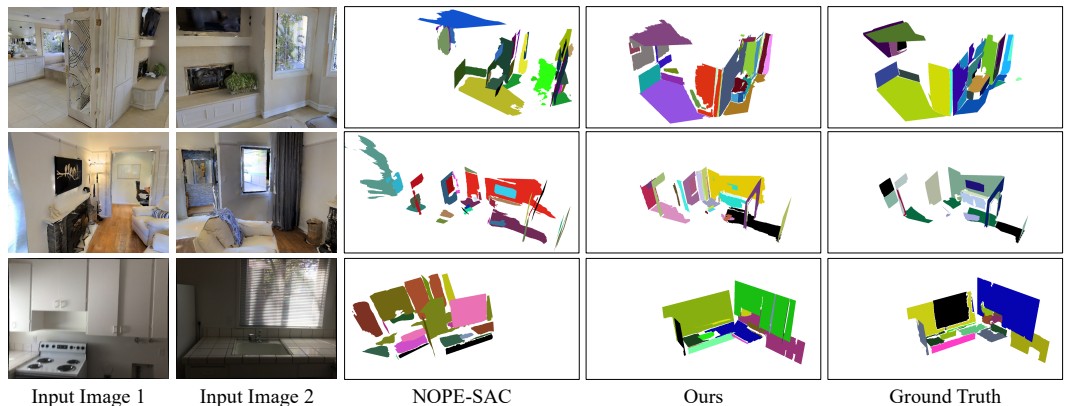

| Input Image 1 | Input Image 2 | NOPE-SAC | Ours | Ground Truth |
| --- | --- | --- | --- | --- |

Figure 5: Comparisons of two-view 3D plane segmentation on the Matterport3D [2] (the first two rows) and the ScanNetV2 [4] (the last row) datasets.

## 4.4 Multi-view Reconstruction with More Than Two Views

PLANA3R currently supports multi-view reconstruction in a pairwise manner, but does not support a single forward pass for inputs with three or more views. Given $N$ input images, we construct $N-1$ image pairs and perform $N-1$ separate forward passes. The planar primitives predicted from each pair are then merged into a common coordinate system. To evaluate this capability, we tested PLANA3R on 50 eight-view samples, sampled every 20 frames from the ScanNetV2 dataset. For a fair comparison, we employed MASt3R also in a pairwise manner as the baseline. The quantitative results of the estimated camera trajectories are summarized in Tab. 4. We provide an example of eight-view reconstruction in Fig. 6.

## 4.5 Alation Study

As introduced in Sec. 3.2 and illustrated in Appendix A.4, to reduce redundancy and achieve a more compact and efficient representation, we design the HPPA module, which regulates the number of planar primitives used for rendering and merging by adjusting the gradient threshold $g_{th}$. In this analysis, we vary the threshold $g_{th}$ such that, for an input resolution of $512 \times 384$, the number of generated primitives ranges from 768 (minimum, $\frac{512}{16} \times \frac{384}{16}$) to 3072 (maximum, $\frac{512}{8} \times \frac{384}{8}$).

By adjusting $g_{th}$, we analyze how the number of per-view primitives affects two-view 3D reconstruction and single-view depth estimation on ScanNetV2 and NYUv2, respectively. As shown in Tab. 5, using approximately half the number of high-resolution primitives achieves performance comparable to using the full high-resolution set. In contrast, relying solely on low-resolution primitives results in a significant drop in accuracy. These results demonstrate that our method can represent scenes efficiently and compactly using a highly sparse set of planar primitives.

Table 4: Comparison with MASt3R on multi-view RRA (Relative Rotation Accuracy) and RTA (Relative Translation Accuracy).

| Method | RRA@5 | RTA@5 | RRA@10 | RTA@10 | RRA@15 | RTA@15 |
|---|---|---|---|---|---|---|
| Ours | 0.9000 | **0.3935** | **0.9985** | **0.7442** | **1.0000** | **0.8614** |
| MASt3R | **0.9828** | 0.2657 | 0.9964 | 0.5371 | **1.0000** | 0.6878 |

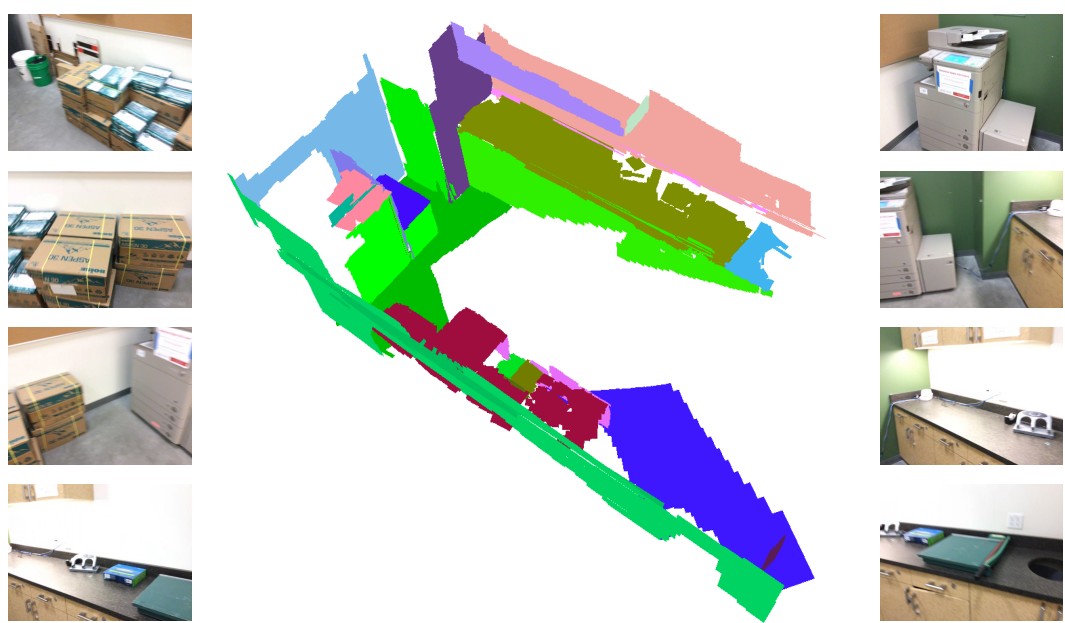

Figure 6: Reconstruction example from 8 frames of a ScanNet sequence. We show the input 8 frames and planar 3D reconstruction.

Table 5: Ablation study on the gradient threshold ($g_{th}$). We show the relationship between the number of per-view primitives and performance.

| Metrics | ScanNetV2 Reconstruction | | | NYUv2 Depth Estimation | | |
|---|---|---|---|---|---|---|
| | Chamfer↓ | F-score↑ | Avg. # primitives | RMSE↓ | $\delta_1$↑ | Avg. # primitives |
| Ours (0) | 0.10 | 93.10 | 3072 | 0.45 | 86.8 | 3072 |
| Ours (0.5) | 0.11 | 92.52 | 1417 | 0.46 | 86.4 | 1565 |
| Ours (10) | 0.11 | 92.32 | 768 | 0.49 | 85.3 | 768 |

## 5 Conclusion

We present PLANA3R, a zero-shot model for metric planar 3D reconstruction that exploits the geometric regularity of indoor scenes using compact 3D planar primitives. Trained without plane annotations, PLANA3R learns geometry priors from large-scale datasets through transformer-based feature extraction and differentiable planar splatting, relying only on depth and normal supervision. Extensive experiments show that PLANA3R generalizes well to out-of-domain indoor scenes and supports efficient planar reconstruction, depth estimation, relative pose estimation, and instance-level plane segmentation from unposed image pairs. We believe that our method establishes a solid foundation for large-scale learning of 3D geometry in indoor scenes and holds great potential for advancing the understanding of indoor environments, with broad applicability in emerging fields such as AR/VR and robotics.

## Acknowledgments

This work was supported by Ant Group Research Intern Program and Ant Group Postdoctoral Program.

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

# A Technical Appendices and Supplementary Material

This appendix provides more details and results of our PLANA3R.

## A.1 Extra Results

We present additional visualization results here.

**Single-view 3D Reconstruction and Plane Segmentation.** To further demonstrate the zero-shot generalization capability of our method in out-of-domain scenes, we evaluate single-view reconstruction and planar segmentation on the 7-Scenes dataset [25]. Although the 7-Scenes dataset is a widely used indoor dataset and is very suitable for out-of-domain evaluation, it does not provide official plane segmentation masks. We attempted to generate them via mesh synthesis but found it challenging to obtain high-quality ground truth labels. Therefore, we present the visualization results. As shown in Fig. 7, our method achieves much better planar segmentation and reconstruction performance than PlaneRecTR.

**Two-view 3D Reconstruction and Plane Segmentation.** We provide more results in Fig. 8 to show that PLANA3R achieves much better performance than NOPE-SAC.

## A.2 Implementation Details

**Data Augmentation.** PLANA3R is trained on a combination of four public indoor datasets as shown in Tab. 7. Due to the low visual quality of rendered RGB images in Habitat [23], we include only a small subset of synthetic images during training and primarily rely on real-world data. Following the training strategy of DUSt3R [33], we apply color jittering and construct symmetric image pairs, $(I^1, I^2)$ and $(I^2, I^1)$, for data augmentation. In our training set (totally around 4M image pairs), we include approximately 0.57M image pairs with no overlap, while the remaining 3.43M pairs are randomly sampled from nearby frames (mainly within the next 10 frames). We observed that incorporating the 0.57M non-overlapping pairs helps the model learn strong geometric priors, enabling it to infer relative pose and scene geometry even when the two views have little or no overlap, as shown in Fig. 8.

Table 6: Results on ScanNetV2 and MP3D (w/o Ext. Data means training without 0.57M non-overlapping image pairs).

| Overlap | Training Data | Translation | | | Rotation | | |
|---|---|---|---|---|---|---|---|
| | | Med. (m)↓ | Mean (m)↓ | ≤0.2m↑ | Med. (°)↓ | Mean (°)↓ | ≤10° ↑ |
| **ScanNetV2** | | | | | | | |
| Easy (>50%) | Full | 0.05 | 0.07 | 97.40 | 1.53 | 1.80 | 99.72 |
| | w/o Ext. Data | 0.05 | 0.09 | 95.73 | 1.77 | 2.77 | 99.16 |
| Medium (15–50%) | Full | 0.06 | 0.10 | 93.56 | 2.02 | 2.68 | 98.88 |
| | w/o Ext. Data | 0.07 | 0.15 | 89.68 | 2.25 | 4.24 | 97.60 |
| Hard (<15%) | Full | 0.12 | 0.26 | 71.98 | 2.68 | 5.53 | 95.67 |
| | w/o Ext. Data | 0.15 | 0.41 | 63.03 | 3.46 | 10.35 | 88.30 |
| Very Hard (non-overlap) | Full | 0.19 | 0.37 | 51.26 | 2.87 | 7.45 | 94.12 |
| | w/o Ext. Data | 0.23 | 0.53 | 46.22 | 4.16 | 16.64 | 79.83 |
| All | Full | 0.07 | 0.13 | 89.16 | 2.01 | 3.16 | 98.30 |
| | w/o Ext. Data | 0.07 | 0.20 | 84.60 | 2.31 | 5.38 | 95.68 |
| **MP3D** | | | | | | | |
| Easy (>50%) | Full | 0.20 | 0.33 | 49.72 | 1.65 | 2.37 | 99.15 |
| | w/o Ext. Data | 0.23 | 0.49 | 44.84 | 1.92 | 4.29 | 97.08 |
| Medium (15–50%) | Full | 0.25 | 0.45 | 37.55 | 2.19 | 4.46 | 96.41 |
| | w/o Ext. Data | 0.29 | 0.66 | 36.75 | 2.41 | 7.67 | 91.81 |
| Hard (<15%) | Full | 0.46 | 0.82 | 17.35 | 2.96 | 11.12 | 85.97 |
| | w/o Ext. Data | 0.59 | 1.03 | 14.26 | 3.17 | 17.84 | 79.52 |
| Very Hard (non-overlap) | Full | 0.82 | 1.11 | 5.63 | 3.00 | 12.37 | 85.28 |
| | w/o Ext. Data | 0.85 | 1.11 | 5.19 | 2.98 | 18.84 | 79.65 |
| All | Full | 0.24 | 0.45 | 40.16 | 2.00 | 4.49 | 96.12 |
| | w/o Ext. Data | 0.29 | 0.64 | 37.14 | 2.24 | 7.62 | 92.37 |

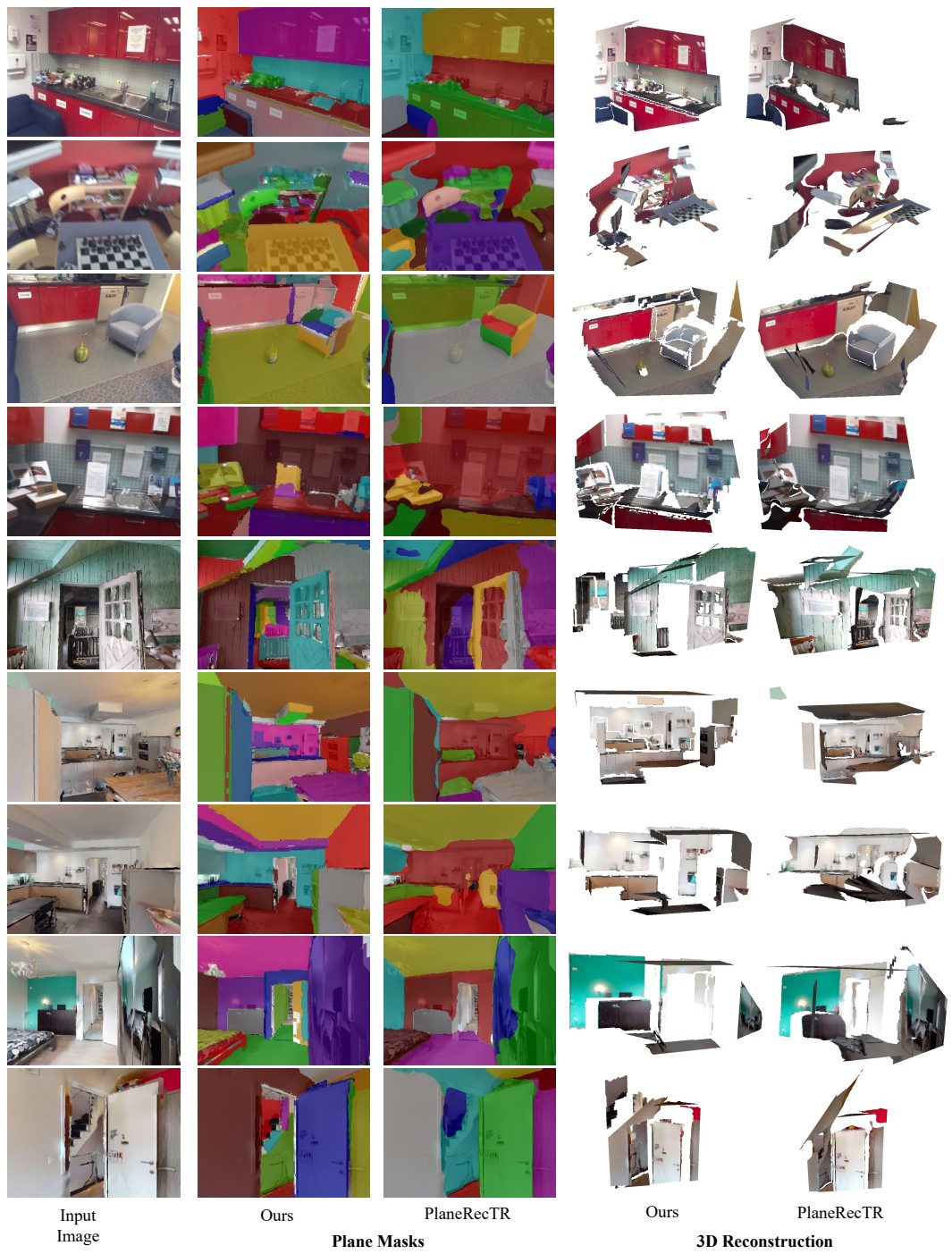

| Input Image | Ours | PlaneRecTR | Ours | PlaneRecTR |
|:---:|:---:|:---:|:---:|:---:|
| | **Plane Masks** | | **3D Reconstruction** | |

Figure 7: Supplement comparisons of single-view plane segmentaion and 3D reconstruction on the 7-Scenes [25] (the first four rows) and Replica [26] (the last four rows).

**Performance Analysis on Pair's Overlap.** We also conduct an additional ablation study to evaluate the impact of incorporating the 0.57M non-overlapping image pairs on model performance during training. In addition, we analyze how the overlap ratio affects performance. Specifically, we define the overlap ratio as the maximum percentage of pixels in one image (either the first or the second) that have direct correspondences in the other. We provide a quantitative analysis to show the impact of including the 0.57M non-overlapping image pairs in Tab. 6.

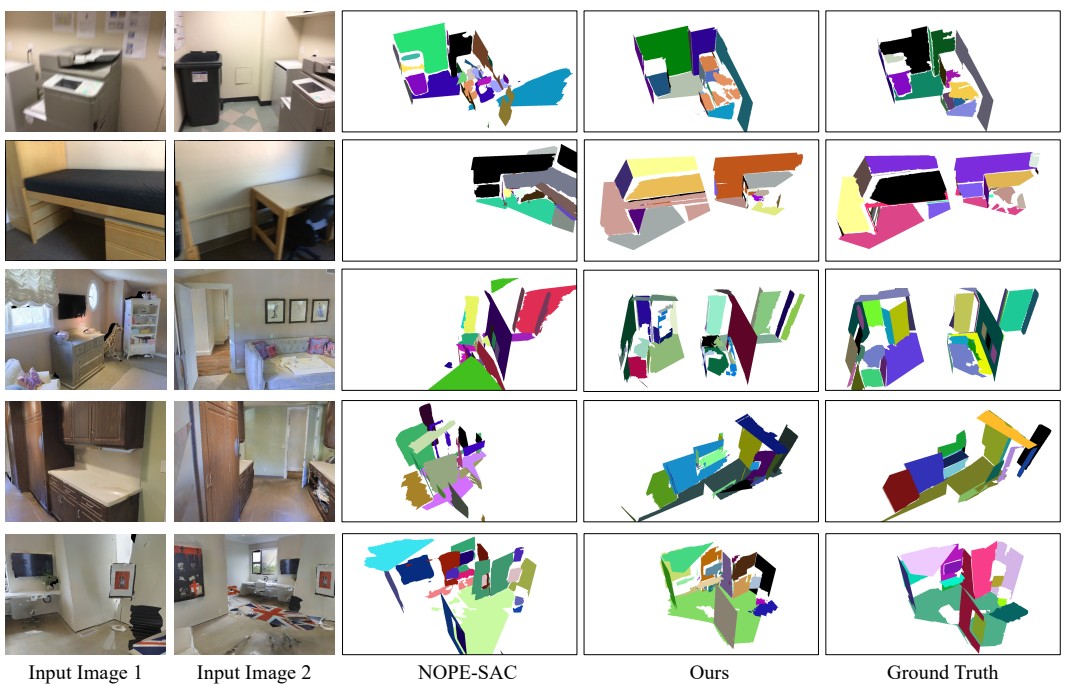

| Input Image 1 | Input Image 2 | NOPE-SAC | Ours | Ground Truth |
|---|---|---|---|---|

Figure 8: Supplement comparisons of two-view 3D plane segmentaion on the ScanNetV2 [4] (the first two rows) and Matterport3D [2] (the last three rows).

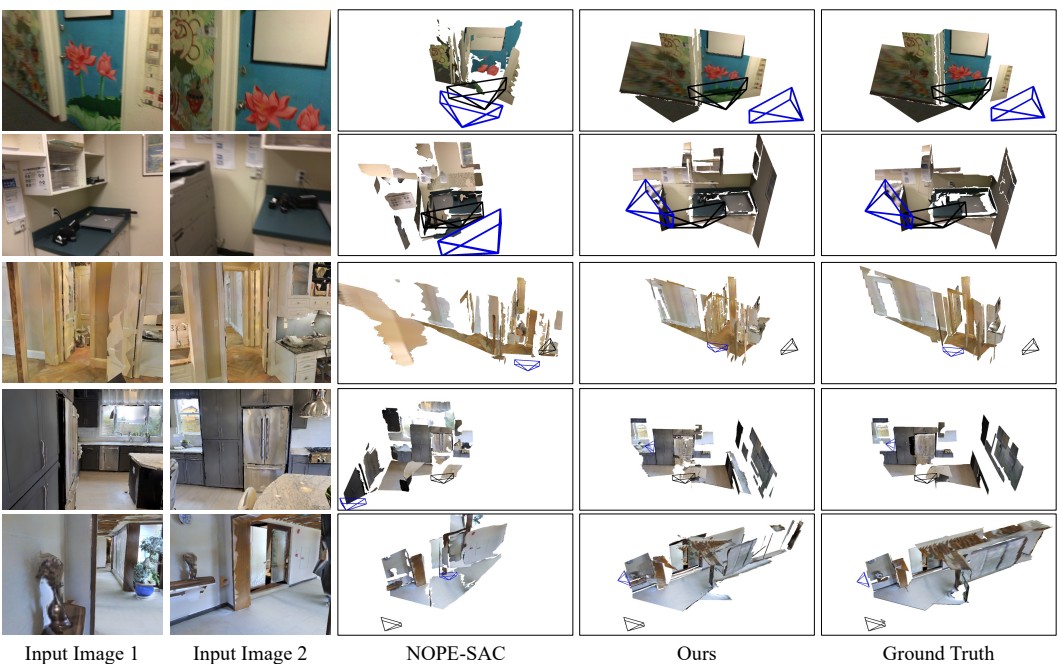

| Input Image 1 | Input Image 2 | NOPE-SAC | Ours | Ground Truth |
|---|---|---|---|---|

Figure 9: Supplement comparisons of two-view 3D plane reconstruction on the ScanNetV2 [4] (the first two rows) and Matterport3D [2] (the last three rows).

As shown in Fig. 11 and Tab. 6, including these non-overlapping pairs improves the model's overall performance. Furthermore, we observe that as the overlap ratio in the test set decreases, the model's accuracy consistently degrades.

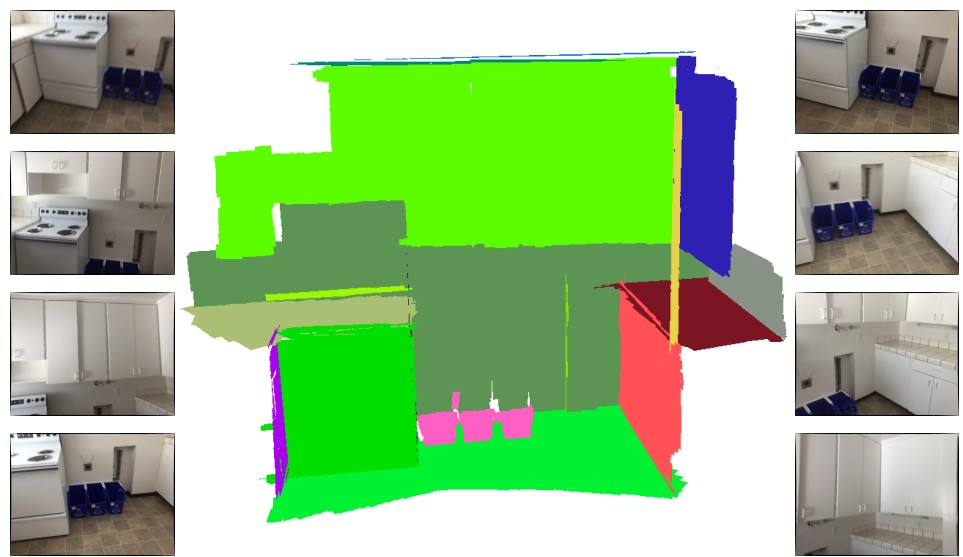

Figure 10: Reconstruction example from 8 frames of a ScanNet sequence. We show the input 8 frames and planar 3D reconstruction.

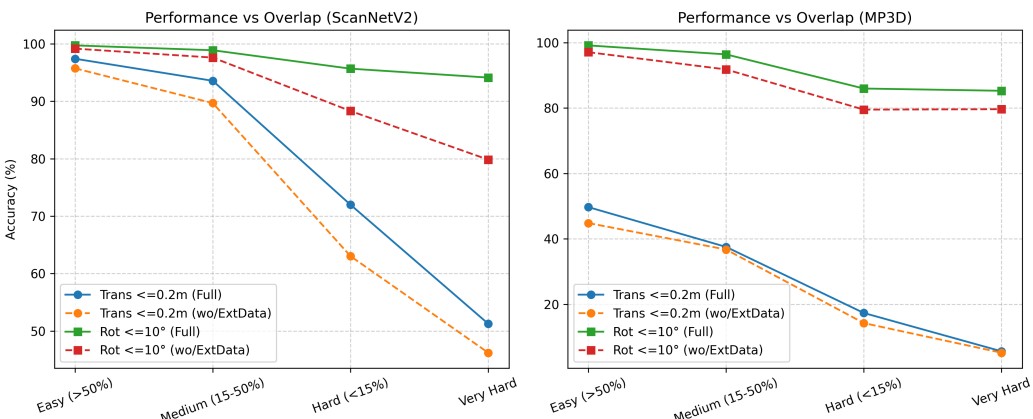

Figure 11: Test-time performance versus overlap degree.

**Plane merge.** Our PLANA3R predicts a set of 3D planar primitives from the inputs. To further achieve instances of 3D planes, we merge the predicted primitives according to their spatial connectivity and geometric similarity. In detail, we greedily merge adjacent planar primitives that meet the merging thresholds (projection distance $\leq 0.1$m and normal angle difference $\leq 25$ degrees) to obtain the final 3D plane instances.

Table 7: Training data and sample sizes for PLANA3R.

| Datasets | Type | N Pairs |
|---|---|---|
| ScanNetV2 [4] | Indoor/Real | 610K |
| ScanNet++ [39] | Indoor/Real | 810K |
| ARKitScenes [5] | Indoor/Real | 2400K |
| Habitat [23] | Indoor/Synthetic | 120k |

**Hyperparameters.** For our final model used for evaluation, we set the loss weights $\alpha_1 = 5$, $\alpha_2 = 5$, $\alpha_3 = 20$ in Eq. (3). We set the loss weights $\beta_1 = 1$, $\beta_2 = 1$, $\beta_3 = 2$ in Eq. (4). We set the loss weights $\gamma_1 = 10$, $\gamma_2 = 10$, $\gamma_3 = 1$ in Eq. (5). We observed that increasing the weights of the patch

loss facilitates faster convergence during the training process. This improvement can be attributed to the fact that more accurate primitive depth estimation ensures proper differentiable rendering and gradient propagation for a larger number of primitives. This occurs because primitives with very incorrect spatial positions or orientations lie outside the camera's viewing frustum, preventing them from being rendered.

## A.3  Runtime Analysis

We evaluate the inference runtime of our PLANA3R using an NVIDIA RTX 3090 GPU. On average, a single feed-forward pass takes 70 ms for predicting planar primitives and relative camera pose. Efficient CUDA-based planar primitive rendering achieves a rate of 1000 fps.

## A.4  Planar Primitive and Primitive Selection

We show the representation of a 3D planar primitive in Fig. 12 (a). We further explain the module of primitive selection in Fig. 12 (b).

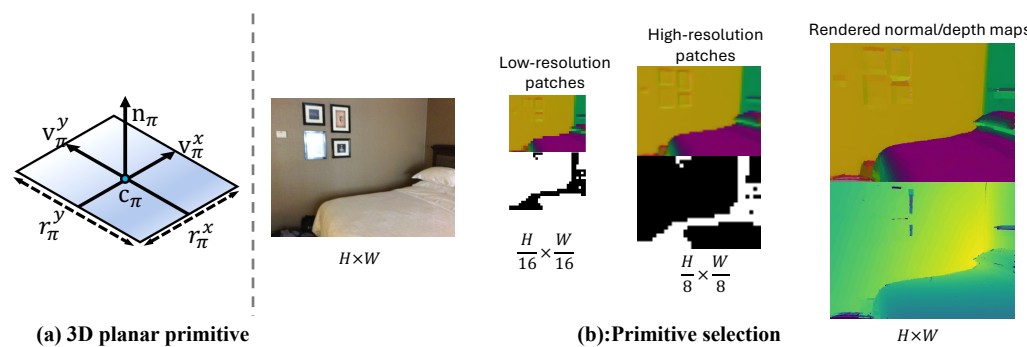

**(a) 3D planar primitive**                **(b):Primitive selection**

Figure 12: (a) A 3D planar primitive using learnable shape parameters. $V_\pi^x$ and $V_\pi^y$ are the positive direction of the X-axis and Y-axis, respectively. (b) We compute the gradient magnitude for each pixel in $\frac{H}{16} \times \frac{W}{16}$ low-resolution predicted normal patches $\mathbf{N}_{\text{low}}^{\text{patch}}$. To combine the low- and high-resolution primitives, we use binary masks (with **white** indicating valid regions) to merge only the valid patches from both resolutions, rather than directly using all predicted primitives.

## A.5  Limitations

We conduct extensive experiments on five test sets. However, when evaluating plane segmentation, the absence of high-quality out-of-domain datasets with reliable plane-level annotations poses a challenge. Although we attempt to generate pseudo-labels, their limited accuracy restricts us to primarily qualitative evaluation. While this represents a limitation in our current analysis, it also highlights the urgent need for better benchmarks in this field. Despite this, the visual results demonstrate the strong potential of our zero-shot method in producing accurate pseudo-plane annotations.

## A.6  Social Impacts

**Positive Social Impact.**   Our zero-shot, pose-free planar primitive framework significantly broadens the applicability of planar 3D reconstruction. Due to its lightweight design and lack of reliance on camera pose or plane annotations, our model can be easily deployed in real-world applications such as AR/VR and robotics, enhancing 3D scene understanding and perception in indoor environments.

**Potential Negative Impact.**   The ease of deployment of our pose-free, zero-shot planar reconstruction model raises potential privacy concerns. Its ability to reconstruct 3D indoor scenes from sparse image pairs could be misused to capture private environments without consent, such as personal residences. While our model performs robustly across various datasets, rare failure cases may occur in challenging scenarios. In real-world deployments, integrating our model into a more comprehensive system is necessary to filter out occasional noisy predictions.

