# OpenReview forum: "PLANA3R: Zero-shot Metric Planar 3D Reconstruction via Feed-forward Planar Splatting"
_NeurIPS.cc/2025/Conference — NeurIPS 2025 poster_

### Official Review · Reviewer_3Lx4 · 2025-06-09

**Clarity:** 3
**Significance:** 3
**Originality:** 3
**Rating:** 5
**Confidence:** 3

**Summary:**

The paper presents a method for indoor metric 3D reconstruction. Given two input views, the method employs Vision Transformers to predict a set of planar primitives as well as the relative camera poses. It uses only depth and normal information for supervision, thereby bypassing the need for 3D annotations and enabling training on multiple large-scale stereo datasets.

**Questions:**

* In the planar reconstruction results, how are the Chamfer distance and F-score calculated using the reconstructed planes? Are points uniformly sampled from the planar surfaces for this evaluation?
* For cases similar to the second row of Figure 7, what does the method predict for the relative camera pose when there is minimal or no overlap between the two input images?
* How are image pairs selected from the datasets to ensure overlap? Or are the image pairs chosen randomly?

**Ethical Concerns:**

["NO or VERY MINOR ethics concerns only"]

**Final Justification:**

The paper presents a method for indoor metric 3D reconstruction using planar primitives and relative pose estimation from two views. The problem is well motivated, as indoor scenes consist of many planar structures, making point-wise representations potentially excessive and less stable. The proposed method is evaluated on large-scale datasets and demonstrates performance that outperforms state-of-the-art point-based methods. The authors have addressed the concerns raised in the reviews, including the influence of view overlap, and have also shown that the method generalizes to a multi-view setup. They have committed to further polishing the final version. Given these reasons, I maintain my initial recommendation for acceptance.

**Limitations:**

It would be beneficial to discuss whether the degree of view overlap poses a challenge for the proposed method.

**Quality:**

4

**Strengths And Weaknesses:**

Strengths:
* The paper is well written and easy to follow.
* The experiments cover many aspects and demonstrate promising results.


Concerns:
* Some evaluation details are not clearly explained in the paper. Specific questions are listed below.
* Since the proposed method focuses on planar reconstruction and includes a plane merging step (Section 3.4), the comparison would be more informative if a simple RANSAC-based plane fitting algorithm were applied to the baselines that output point clouds.
* In two-view reconstruction problems, one might assume there is some overlapping area between the views. It would be helpful to discuss how the overlap ratio affects performance. Given that the method is trained on indoor datasets, it would also be interesting to see if these datasets contain strong priors that allow the model to "guess" the relative pose and geometry even when the two views have little or no overlap.

---

> ### Author Rebuttal · Authors · 2025-07-29
>
> We sincerely thank you for your positive feedback on our manuscript, especially your recognition of our model’s strong performance and comprehensive experiments. We greatly appreciate your constructive comments and address your questions in detail below.
>
> ### 1. On the Chamfer distance and F-score calculation
> Your understanding is correct, we uniformly sample points from the planar surfaces for this evaluation.
>
> ### 2. Comparison with RANSAC-based plane fitting baseline
> We supplement a new RANSAC-based plane fitting baseline here. Our method achieves much better performance.
> | Method         | ScanNetV2               |                       | Matterport3D            |                       |
> |----------------|-------------------------|-----------------------|--------------------------|-----------------------|
> |                | Chamfer ↓               | F-score ↑             | Chamfer ↓                | F-score ↑             |
> | Mast3R         | 0.21                    | 74.92                 | 0.49                     | 30.01                 |
> | Mast3R+RANSAC  | 0.20                    | 73.12                 | 0.49                     | 28.35                 |
> | Ours           | **0.11**                | **92.52**             | **0.32**                 | **56.63**             |
>
> ### 3. How we sample pairs as training set
> Your understanding is correct—thank you for the insightful observation. In our training set (totally 4M image pairs),  we include approximately 0.57M image pairs with no overlap, while the remaining 3.43M pairs are randomly sampled from nearby frames (mainly within the next 10 frames). We observed that incorporating the 0.57M non-overlapping pairs helps the model learn strong geometric priors, enabling it to infer relative pose and scene geometry even when the two views have little or no overlap. Below, we provide a quantitative analysis to show the impact of including the 0.57M non-overlapping image pairs.
>
> ### 4. Performance analysis on pair’s overlap
> We conduct an additional ablation study to evaluate the impact of incorporating the 0.57M non-overlapping image pairs on model performance during training. In addition, we analyze how the overlap ratio affects performance. Specifically, we define the overlap ratio as the maximum percentage of pixels in one image (either the first or the second) that have direct correspondences in the other.
>
> As shown in the table below, including these non-overlapping pairs significantly improves the model’s overall performance. Furthermore, we observe that as the overlap ratio in the test set decreases, the model’s accuracy consistently degrades.
>
> **ScannetV2** (wo/ExtData means training without 0.57M non-overlapping image pairs):
> | Overlap                |Training Data| Translation |  |   |   | Rotation|  |   |
> | ---------------------- |-------------| ------- | ------- | ------ | - | --------- | --------- | ----- |
> |                 || Med.(m)↓ | Mean(m)↓ | <=0.2m↑  |   | Med.(deg) ↓| Mean(deg)↓ | <=10°↑  |
> | Easy (>50%)            |Full| 0.05    | 0.07    | 97.40  |   | 1.53      | 1.80      | 99.72 |
> |                                 | wo/ExtData            | 0.05    | 0.09    | 95.73  |   | 1.77      | 2.77      | 99.16 |
> | Medium (15%-50%)     |Full  | 0.06    | 0.10    | 93.56  |   | 2.02      | 2.68      | 98.88 |
> | | wo/ExtData            | 0.07    | 0.15    | 89.68  |   | 2.25      | 4.24      | 97.60 |
> | Hard (<15%) |Full           | 0.12    | 0.26    | 71.98  |   | 2.68      | 5.53      | 95.67 |
> ||wo/ExtData            | 0.15    | 0.41    | 63.03  |   | 3.46      | 10.35     | 88.30 |
> | Very Hard (non-overlap)| Full | 0.19    | 0.37    | 51.26  |   | 2.87      | 7.45      | 94.12 |
> | |wo/ExtData            | 0.23    | 0.53    | 46.22  |   | 4.16      | 16.64     | 79.83 |
> | All | Full                   | 0.07    | 0.13    | 89.16  |   | 2.01      | 3.16      | 98.30 |
> |     |wo/ExtData            | 0.07    | 0.20    | 84.60  |   | 2.31      | 5.38      | 95.68 |
>
> -----------------------------------------------------------------------------------------
>
> **MP3D**  (wo/ExtData means training without 0.57M non-overlapping image pairs):
> | Overlap                |Training Data| Translation |  |   |   | Rotation|  |   |
> | ---------------------- |-------------| ------- | ------- | ------ | - | --------- | --------- | ----- |
> |                 || Med.(m)↓ | Mean(m)↓ | <=0.2m↑  |   | Med.(deg) ↓| Mean(deg)↓ | <=10°↑  |
> | Easy (>50%)            | Full        | 0.20    | 0.33    | 49.72  |   | 1.65      | 2.37      | 99.15 |
> |                        | wo/ExtData  | 0.23    | 0.49    | 44.84  |   | 1.92      | 4.29      | 97.08 |
> | Medium (15%-50%)       | Full        | 0.25    | 0.45    | 37.55  |   | 2.19      | 4.46      | 96.41 |
> |                        | wo/ExtData            | 0.29    | 0.66    | 36.75  |   | 2.41      | 7.67      | 91.81 |
> | Hard (<15%)            | Full        | 0.46    | 0.82    | 17.35  |   | 2.96      | 11.12     | 85.97 |
> |                        | wo/ExtData            | 0.59    | 1.03    | 14.25  |   | 3.18      | 17.84     | 79.23 |
> | Very Hard (non-overlap) | Full        | 0.82    | 1.11    | 5.63   |   | 3.00      | 12.37     | 85.28 |
> |                        | wo/ExtData            | 0.85    | 1.11    | 5.19   |   | 2.98      | 18.84     | 79.65 |
> | All                    | Full                  | 0.24    | 0.45    | 40.16  |   | 2.00      | 4.49      | 96.12 |
> |                        | wo/ExtData            | 0.29    | 0.64    | 37.14  |   | 2.24      | 7.62      | 92.37 |
>
> We hope these clarifications and supplement results effectively address your feedback. Thank you again for your valuable insights, which have significantly strengthened our manuscript.

---

> > ### Comment · Reviewer_3Lx4 · 2025-08-01
> >
> > Thanks for the clarification and additional tables.
> > A small suggestion: it could be helpful to include a plot showing test-time performance versus the degree of overlap, either in the main paper or the supplementary material, to better illustrate and discuss the method’s limitations.
> > I do not have further questions and I will keep my initial rating.

---

> > > ### Author Response · Authors · 2025-08-02
> > >
> > > Thank you for the suggestion. We will include a plot of test-time performance versus overlap degree in the supplementary material to better illustrate the behavior of our method.

---

### Official Review · Reviewer_CYmQ · 2025-06-22

**Clarity:** 3
**Significance:** 2
**Originality:** 2
**Rating:** 3
**Confidence:** 3

**Summary:**

The paper proposes PLANA3R, a feed-forward, pose-free framework for metric-scale planar 3D reconstruction from unposed two-view images. It uses sparse planar primitives and learns from depth and normal maps without plane annotations and shows strong zero-shot generalization across indoor datasets.

**Questions:**

1. Can the model be extended to handle more than two input images? Have you experimented with applying the framework to reconstructions using a different number of input views?
2. Could the authors provide a quantitative analysis of the computational performance achieved by the hierarchical feature design, including measurements of runtime, memory consumption, or floating point operations (FLOPs)?
3. How much resource consumption and calculation speed increase does the design of image texture mask bring?
4. Are there typical patterns or scenarios where the hierarchical design fails, e.g., in low-texture areas or complex clutter?

**Ethical Concerns:**

["NO or VERY MINOR ethics concerns only"]

**Final Justification:**

Thank authors for their very detailed responses and additional experiments. My concerns have been resolved by the responses. I decide to raise my score accordingly.

**Limitations:**

The authors have adequately addressed the limitations and societal impact in the main paper.

**Paper Formatting Concerns:**

No formatting issues were identified.

**Quality:**

2

**Strengths And Weaknesses:**

Strengthens：
- The use of planar primitives offers a concise and interpretable alternative to dense point clouds.
- Extensive experiments demonstrate state-of-the-art performance across different tasks.
- The approach trains without explicit plane-level supervision, relying only on depth and normal maps, which are more readily available.
- PLANA3R predicts relative pose and 3D structure in a single forward pass, making it suitable for real-time applications.

Weaknesses：
- The method feels like a relatively direct combination of ideas from PlanarSplatting and DUSt3R/MASt3R, with limited conceptual novelty.
- It is unclear how well it generalizes to multi-view or more complex scenes beyond pairwise settings.
- The hierarchical prediction of 16× and 8× downsampled features constitutes a core design intended to select appropriate fine-grained textures in different regions to reduce computational overhead and improve inference speed. However, the authors have not yet provided quantitative evidence, such as measurements of runtime, memory consumption, or floating-point operations (FLOPs). Moreover, although the design is intended to accelerate computation, it requires computing gradient-based texture masks, which itself incurs additional computational cost. It remains unclear whether this overhead offsets the claimed improvement in computational speed.

---

> ### Author Rebuttal · Authors · 2025-07-30
>
> We sincerely thank the reviewers for recognizing the training strategy and application value of our Plana3r method, which enables end-to-end real-time reconstruction and pose estimation without requiring plane annotations. Below, we address your concerns with detailed clarifications and planned revisions.
>
> ### 1. On the Conceptual Novelty of Plana3r
>
> The feedforward modeling of multi-view geometry (MVG) gained traction with the introduction of Dust3r/Mast3r in early 2024, followed by other “3R” family methods addressing longstanding challenges in classical MVG through point map representations and large vision transformers, including but not limited to multi-view aggregation (VGGT [1]) and dynamic scene reconstruction (Cut3r [2]). While these methods have improved 3D reconstruction performance, **point map representations are often redundant and inefficient**. All existing 3R methods must output a large number of 3D points to characterize a scene. Thus, pursuing compact and efficient representations in feedforward 3D reconstruction is both urgent and impactful.
>
> Plana3r addresses this by introducing a compact planar 3D representation into the 3R framework, integrating the strengths of feedforward modeling and planar primitives. We build upon PlanarSplatting, which provides an efficient rendering scheme from planar primitives. However, unlike PlanarSplatting, our method does not require precomputed camera poses. From that perspective, Plana3r is a non-trivial technical advancement. Our key contributions are as follows:
>
> * **Sparse Planar Representation**: Unlike Dust3r/Mast3r, which use a DPT head for dense point regression, and PlanarSplatting, which requires dense depth maps for primitive initialization, Plana3r directly regresses sparse planar primitives from \$H/16 \times W/16\$ patches. Our novel HPPA module adaptively adjusts the number of primitives, ensuring a sparse yet sufficient representation that captures scene geometry. This reduces storage demands while preserving detail in textureless regions. **Devising a new 3D vision foundation model using the planar representation has unique application value**.
>
> * **Metric Reconstruction and Pose Estimation**: Unlike most related work, Plana3r supports metric reconstruction and pose estimation. Leveraging the good data distribution of indoor data, our model achieves high data utilization and accurate metric estimation, offering new insights into 3D vision foundation model design.
>
> * **Optimized Training Strategy**: Training with differentiable rendering in PlanarSplatting presents challenges due to the randomness in initialization at the start of training—many primitives are initially placed outside the camera’s view, leading to unstable gradients. To mitigate this, we adopt a two-phase training strategy designed to stabilize the training.
>
> ### 2. On the motivation of Hierarchical Primitive Prediction Architecture (HPPA)
> We believe there may be a slight misunderstanding regarding the motivation behind **HPPA**. As stated in our paper (Section 3.1, line 143 to line 145, Page 4):
> > *“To balance efficiency and representational fidelity, we propose a hierarchical primitive prediction architecture (HPPA) to fit the scene using planar primitives, enabling compact and efficient modeling of scene geometry with sparse primitives.”*
>
> Our primary goal with HPPA is to achieve **a more compact and efficient geometric representation using fewer primitives**. Here, the term **"efficiency"** specifically refers to the **number of primitives required to represent a scene**, rather than the inference runtime. While we do not claim that HPPA improves inference speed, we do provide a runtime and FLOPs breakdown to demonstrate that the **additional computational overhead introduced by the image texture mask is minimal**. As the calculations are conducted on $H/16 \times W/16$ and $H/8 \times W/8$ patches, the extra cost is negligible in practice. If the term **“efficiency”** has caused any confusion, we will revise the wording in **Section 3.1 and Section 3.2** to clarify that it refers to **representational efficiency (i.e., fewer primitives)** rather than runtime efficiency. Below is the detailed runtime analysis tested on NVIDIA RTX 3090:
>
> ```
> encoder:               12.1 ms
> decoder:               32.8 ms
> upsample:               0.4 ms
> head_prediction:       18.0 ms
> hppa_low_view1:         2.0 ms
> hppa_high_view1:        1.5 ms
> hppa_low_view2:         1.0 ms
> hppa_high_view2:        0.9 ms
> rendering:              0.9 ms
> -------------------------------
> Total HPPA overhead:    around 5.0 ms
> Total compute:         ~1.54 TFLOPs
> Memory footprint:      ~7.8 GB
> Model params: 540M (DUSt3R is 571M, MASt3R is 689M)
> ```
>
> ### 3. On the multi-view  reconstruction with more than two input images
> Plana3r currently supports multi-view reconstruction in a pairwise manner, but does not support a single forward pass for inputs with three or more views. Given $N$ input images, we construct $N-1$ image pairs and perform $N-1$ separate forward passes. The planar primitives predicted from each pair are then merged into a common coordinate system. To evaluate this capability, we tested Plana3r on 50 eight-view samples, sampled every 20 frames from the ScanNetV2 dataset. For fair comparison, we employed MASt3R also in a pairwise manner as the baseline. The quantitative results of the estimated camera trajectories are summarized below:
>
> RRA (Relative Rotation Accuracy) and RTA (Relative Translation Accuracy)
> | Metric  ↑ | Ours    | mast3r |
> |----------|--------|-----------|
> | RRA@5     | 0.9000 |  **0.9828**    |
> | RTA@5    | **0.3935** | 0.2657    |
> | RRA@10   | **0.9985** |  0.9964    |
> | RTA@10   | **0.7442** | 0.5371    |
> | RRA@15   | **1.0000** | **1.0000**    |
> | RTA@15   | **0.8614**    | 0.6878    |
>
> This paper focuses on two-view reconstruction and relative pose estimation—core problems in 3D vision with broad applicability. Our key contribution lies in designing a 3D vision foundation model with a new representation and proposing a new perspective for end-to-end metric indoor scene reconstruction. Currently, we train with two-view inputs at \$512 \times 384\$ resolution, which already imposes high GPU demands (Section 4.1). While we have not yet conducted end-to-end multi-view training, Plana3r supports multi-view reconstruction at inference time with multiple feed-forwards. Extending to full multi-view training is straightforward—e.g., by scaling the data and following approaches like VGGT [1]. We appreciate the suggestion and agree that one-pass multi-view reconstruction is a promising direction. We plan to explore this when we have sufficient computational resources in the future.
>
> ### 4. Failure case analysis:
> As shown in Figures 3, 4, 5, 7, and 8, our method will not fail in low-texture areas and can yield good plane segmentation and planar reconstruction. For the complex clutter scene, if there are many non-planar areas and very complex textures in the scene, such as potted plants, etc., it is difficult for our method to achieve high-quality reconstruction. However, the motivation of this paper is for the man-made, structured indoor scene.
>
> [1] VGGT: Visual Geometry Grounded Transformer (CVPR 2025)
>
> [2] Continuous 3D Perception Model with Persistent State (CVPR 2025)
>
> We hope these clarifications address your concerns. Thank you again for your valuable feedback, which has strengthened our manuscript.

---

> ### Author Response · Authors · 2025-08-05
> **Follow-up post**
>
> Dear Reviewer CYmQ,
>
> We sincerely appreciate your thoughtful comments on our paper.
>
> As the reviewer-author discussion period will conclude on August 6, we would like to kindly check if there are any remaining concerns that were not fully addressed in our rebuttal. If you have any additional feedback or suggestions regarding our submission, we would be very grateful to receive them.
>
> Thank you again for your time and consideration. We truly value the review process and your input in helping us improve the quality of our work.
>
> Best regards,
>
> PLANA3R Team

---

> > ### Comment · Reviewer_CYmQ · 2025-08-08
> > **Response to rebuttal**
> >
> > Thank you for your rebuttal. My concerns have been fully addressed, and I will revise my score accordingly.

---

### Official Review · Reviewer_w5iN · 2025-06-28

**Clarity:** 3
**Significance:** 3
**Originality:** 3
**Rating:** 5
**Confidence:** 5

**Summary:**

This paper proposes PLANA3R, a strong method that extends the 3R framework to handle structured indoor scenes. The key idea is to pretrain a feed-forward plane prediction network that learns to generate sparse planar primitives in metric scale, using only depth and normal maps as supervision from large indoor datasets. The differentiable PlanarSplatting module plays a key role for end-to-end training of the pipeline. PLANA3R predicts planar 3D primitives directly from two unposed images, supporting multiple downstream tasks such as depth estimation, plane segmentation, and reconstruction. Extensive experiments demonstrate that PLANA3R achieves impressive performance, showing clear improvements over prior methods in both qualitative visualizations and quantitative metrics.

**Questions:**

1.	There are many incomplete depths in GT datasets because they are obtained from sensored LiDAR cameras. How do you deal with the incompleteness in GT depths?
2.	PLANA3R uses estimated normal from Metric3Dv2 in scenarios where there is no GT normal. However, we can also generate pseudo normal from GT depths just as what 2DGS does. It may provide more accurate normal supervision. Does the author observe any difference in quality or training stability between these two pseudo normals?
3.	Why are the input images in Figure 3,4,5 with such low quality? Aren’t they come from GT image sequences?

**Ethical Concerns:**

["NO or VERY MINOR ethics concerns only"]

**Final Justification:**

All of my concerns have been addressed. I would keep my rating of accept.

**Limitations:**

No evident limitations.

**Quality:**

4

**Strengths And Weaknesses:**

Strengths

1. This paper is an interesting and meaningful extension of the 3R framework towards structured planar 3D reconstruction. By leveraging PlanarSplatting as a differentiable renderer, the entire training pipeline is made fully end-to-end trainable without explicit plane-level annotations, which is a notable technical strength.
2. The proposed hierarchical primitive prediction combines low-resolution and high-resolution primitives to achieve a balance between primitive numbers and reconstruction fidelity, which is novel and practically effective.
3. The experimental setup is thorough, covering diverse datasets and multiple tasks, with strong improvements over baselines in both quantitative metrics and visual quality.
4. The paper is well-written, the technical presentation is clear, and the results are convincingly demonstrated.

Weaknesses

I did not find any major weaknesses. One minor point: I suggest the author to reorganize Figure 2 to more clearly illustrate how the low-resolution and high-resolution primitives are generated. As currently drawn, it may leave readers with the impression that the high-res primitives are simply upsampled directly from the low-res ones, which is not the case.

---

> ### Author Rebuttal · Authors · 2025-07-30
>
> We sincerely thank you for your positive feedback on our paper. We are especially grateful that you found the research topic intriguing and recognized the novelty and technical strengths of our proposed method. Your insightful comments have helped us refine our work further. Below, we provide detailed responses to your suggestions and questions, along with the revisions we plan to make.
>
> ### 1. On framework figure
> We appreciate your suggestion to enhance Figure 2 and the descriptions of high-resolution and low-resolution primitives. In the revised manuscript, we will update Figure 2 to better illustrate these concepts and provide more detailed explanations of high-resolution and low-resolution primitives in the manuscript to improve clarity and accessibility.
>
> ### 2. On sensor ground-truth depth
> We acknowledge that the sensor-provided ground-truth depth is relatively sparse. During training, we apply masks to filter out invalid values (e.g., NaNs) in the depth data. These masked regions are excluded from the loss computations in Equations (3) and (4).
>
> ### 3. On pseudo ground-truth normals
> Thank you for raising this insightful point. Following your suggestion, we generated pseudo ground-truth normals from the sensor depth data. Compared to the normals generated by Metric3Dv2, these pseudo normals contain more invalid regions and exhibit less smoothness. To evaluate their impact, we trained a version of our model using these pseudo normals as supervision, as shown in the table below. It can be seen that the model performs better when using the normal provided by Metric3Dv2. Interestingly, using a poor normal not only reduces the performance of 3D reconstruction but also the accuracy of pose estimation.
>
> **ScanNetV2 dataset**
> | Method                        |        Pose Translation        |                      |  Pose Rotation           |                      | Reconstruction |               |
> |------------------------------|--------------------|----------------------|----------------------|----------------------|----------------|---------------|
> |                              | Med. (m)↓           | Mean (m)↓             | Med.(deg)↓            | Mean(deg)↓            | Chamfer ↓      | F-score ↑     |
> | normal_from_depth            | 0.11  | 0.18  | 4.22      | 5.78      | 0.22       | 73.18      |
> | Full (normal from metric3d)  | **0.07**  |  **0.13**  |  **2.01**      |  **3.16**      | **0.11**       |  **92.52**      |
>
> **MP3D dataset**
>
> | Method                        |        Pose Translation       |                      |        Pose Rotation               |                      | Reconstruction |               |
> |------------------------------|--------------------|----------------------|----------------------|----------------------|----------------|---------------|
> |                              | Med. (m)↓           | Mean (m)↓             | Med.(deg)↓            | Mean(deg)↓            | Chamfer ↓      | F-score ↑     |
> | normal_from_depth            | 0.28  | 0.47  | 4.32      | 6.59      | 0.34       | 49.13      |
> | Full (normal from metric3d)  |  **0.24**  |  **0.45**  |  **2.00**     |  **4.49**      |  **0.32**       |  **56.63**      |
>
> ### 4. On MP3D RGB images
> The RGB images in the MP3D dataset are rendered from the officially provided mesh, and thus exhibit lower quality and certain distortions compared to real camera-captured images. Nonetheless, these inputs further underscore the strong generalization capability of our Plana3r model. This result highlights the effectiveness of planar primitives as a robust intermediate representation: despite the degraded image quality, Plana3r consistently predicts high-quality planar primitives, enabling accurate reconstruction of planar surfaces.
>
> We hope these clarifications and planned revisions effectively address your feedback. Thank you again for your valuable insights, which have significantly strengthened our manuscript.

---

> > ### Comment · Reviewer_w5iN · 2025-08-01
> > **Response to the rebuttal**
> >
> > Thanks for the rebuttal. All of my concerns have been addressed. I would keep my rating of accept.

---

### Official Review · Reviewer_9yaB · 2025-07-02

**Clarity:** 2
**Significance:** 3
**Originality:** 3
**Rating:** 4
**Confidence:** 3

**Summary:**

This paper proposes PLANA3R, a two-view metric planar reconstruction method designed for indoor environments. Built upon the DUSt3R network framework, the authors introduce network heads that output multi-level planar primitives and utilize normal gradients to guide the selection of primitives, thereby reducing the number of primitives. Additionally, the method leverages normal and depth images from the dataset to supervise training, without requiring ground-truth plane annotations. Experimental results demonstrate that PLANA3R outperforms the baselines on tasks such as 3D surface reconstruction, depth estimation, and relative pose estimation.

**Questions:**

see Weaknesses.

**Ethical Concerns:**

["NO or VERY MINOR ethics concerns only"]

**Final Justification:**

The authors have responded to my concerns appropriately and professionally. Given the impressive results demonstrated in this work, I have decided to raise my score to Borderline accept.

**Limitations:**

Yes, in the supplementary.

**Quality:**

3

**Strengths And Weaknesses:**

**Strengths:**
1. The motivation of the paper is clear: planar primitives are well-suited for representing lightweight, structured indoor environments. The Introduction and Related Work sections are logically organized and easy to follow.
2. The use of multi-level, multi-resolution primitive outputs combined with a normal gradient-based selection mechanism effectively reduces the number of primitives.
3. Experimental results demonstrate that the proposed method delivers strong performance across multiple tasks.

**Weaknesses:**
1. Since PlanarSplatting is a fundamental component of this work, it would be helpful to include a more detailed explanation of its core ideas to make the paper more self-contained and accessible to readers unfamiliar with it.
2. The Method section lacks clarity in several important aspects, making it difficult to follow. For example, in Equation (1), the meaning of the indices $i$ and $j$ in $d_{\pi}^{i,j}$, $\mathbf{r}{\pi}^{i,j}$, and $\mathbf{q}{\pi}^{i,j}$ is not clearly explained. Does $\mathbf{q}{\pi}^{i,j}$ represent the quaternion of a primitive generated from image $i$, expressed in the coordinate frame of camera $j$? or $\mathbf{q}{\pi}^{i,j}$ is simply under its camera coordinate system.  The lack of clarity regarding the coordinate systems involved makes it difficult to fully understand how the training losses are formulated and applied.
3. In the *Training losses and training strategies* section, "As a result, directly applying the differentiable planar primitive rendering ......."
Actually, I don't understand what the rendering is used for. I couldn't find any information that explained it. Do you mean render the depth and normal images from image $i$ to camera $j$, or render images from all the primitives under the same coordinate system (use $q$)? And then use the rendered image to compare it with the ground truth? What's the target for introducing this rendering loss to the training stage?
4. It is also not clear whether the warm-up training phase includes the pose loss.

Overall, I appreciate the value of this work and its strong performance across different tasks. However, the lack of clarity in important implementation details makes it difficult to fully understand how the proposed method is realized. If the authors can provide more detailed explanations regarding these aspects, I would be willing to consider increasing my score.

---

> ### Author Rebuttal · Authors · 2025-07-29
>
> We sincerely thank you for your thorough and insightful feedback, which has significantly enhanced the quality of our manuscript. We greatly appreciate the recognition of the value and performance of our approach. Below, we address your suggestions and questions with detailed clarifications and outline revisions.
>
> ### 1. On PlanarSplatting introduction
> Thank you very much for your suggestion. We will include an introduction to PlanarSplatting in our revised manuscript to improve clarity and make the paper more self-contained. Specifically, we will add the following description in Section 3.1:
>
> >'' PlanarSplatting's rasterization is a core component of our Plana3r model. PlanarSplatting is an ultra-fast and accurate method for reconstructing planar surfaces in indoor scenes from multi-view images. Rather than detecting, matching, or tracking planes in 2D or 3D, it directly operates on a set of 3D planar primitives, "splatting" them into **dense depth and normal maps** via differentiable, CUDA-accelerated rasterization.''
>
> ### 2. Symbol definitions and coordinate frames
>
> We apologize for the insufficient explanation of symbols in the manuscript. Your interpretation of $q_{\pi}^{i,j}$ is correct—it denotes the quaternion of a planar primitive generated from image $I^i$, represented in the coordinate frame of camera $j$. In our training setup, **we fix $j = 1$, treating $I^1$ as the reference frame**. Accordingly, the ground-truth camera pose of $I^1$ is set to the $4{\times}4$ identity matrix.  The notation $r_{\pi}^{i,j}$ refers to the radii array of planar primitives associated with image $I^i$. Since the radii represent side lengths and are independent of camera $j$, this notation may be misleading. We will revise it to $r_{\pi}^{i}$ for clarity. Similarly, $d_{\pi}^{i,j}$ was a typo and should be corrected to $d_{\pi}^{i}$, which denotes the depth map of the planar primitive centers associated with image $I^i$. We will revise the Equation (1) in manuscript to clarify the notation and coordinate system accordingly as:
> > Input: \$(I^i, K^i)_{i=1,2}\$
>
> > Output of Plana3r: $(d_{\pi}^{i},\ r_{\pi}^{i},\ q_{\pi}^{i,j})$, $j{=}1$, and $P_{\text{rel}}$.
>
> Then, the primitive centers $c_{\pi}$ from both $I^1$ and $I^2$ are transformed into the coordinate frame of camera $j=1$ using the corresponding intrinsics $K^i$, depthmap $d_{\pi}^{i}$ and camera poses (page 4, line 127 to line 128). Therefore, throughout training, the primitive centers $c_{\pi}$, quaternions, and camera poses of both $I^1$ and $I^2$ are consistently expressed in the coordinate frame of camera $j = 1$.
>
> ### 3. Differentiable planar primitive rendering in training stage
>
> We appreciate the reviewer’s question regarding the phrase: *“As a result, directly applying the differentiable planar primitive rendering...”* Here we provide further clarification:
>
>  **- What to render:**
>
> Our **Plana3r** model takes as input two unposed images and their camera intrinsics, and outputs a set of planar primitives in the coordinate frame of the camera \$j{=}1\$. During training, we use the differentiable rasterizer from PlanarSplatting to render dense depth and normal maps **at the original resolution (\$H \times W\$)** with sparse primitives predicted from low-resolution image features (Page 15, Figure 9 (b)).
>
> * For \$I^1\$, rendering is done using the \$4{\times}4\$ identity matrix as its camera pose.
> * For \$I^2\$, rendering is done using the ground-truth relative pose from \$I^2\$ to \$I^1\$.
>
> The rendered depth \$D^{\text{render}}\$ and normal \$N^{\text{render}}\$ for each image are computed in their respective camera coordinate systems. Thus, while the inputs to the renderer (camera poses and primitives) are defined in the coordinate frame of \$I^1\$, the rendered outputs (depth and normal maps) are expressed in each image’s own coordinate frame.
>
> **- Target of rendering loss:**
>
> Note that the primitives are initially predicted from low-resolution image patches and are supervised using patch-level losses at resolutions of ($H/16 \times W/16$) and ($H/8 \times W/8$), as described in Equation (3).  **This patch-level loss does not use the radii information of planar primitives, but only supervises the depth and quaternion of predicted planar primitives**. To capture accurate geometry of the scene and let the gradient flow optimize the radii of planar primitives, we further supervise the model by computing losses between the rendered and ground-truth depth and normal maps (as shown in Equation (4)) using PlanarSplatting's rasterization, at the original image size (\$H \times W\$). This encourages the model to **predict accurate primitive shapes (radii)**, depths, and orientations, thereby improving the geometric fidelity of the reconstructed scene. We will incorporate this clarification into the manuscript.
>
> ### 4. On warm-up training phase
> Thank you for highlighting the need for clarity regarding the warm-up training phase. As noted in Section 4.1 (page 6, line 205 to line 206), the warm-up phase includes the pose loss. We will ensure this is more prominently explained in the revised manuscript to avoid any oversight.
>
> We hope these clarifications address your questions effectively. Thank you again for your valuable reviews.

---

> > ### Comment · Reviewer_9yaB · 2025-08-04
> >
> > Thanks for the detailed rebuttal. It has addressed my concerns, and I will revise my score accordingly.

---

### Decision · Program_Chairs · 2025-09-17

**Decision:**

Accept (poster)

**Comment:**

The paper introduces a pose-free framework for metric-scale planar 3D reconstruction from unposed two-view images, leveraging sparse planar primitives and supervision from depth and normal maps without requiring plane annotations. Reviewer’s feedback was largely positive, noting the paper’s strengths while requesting clarification on several points, including (1) whether the model can handle more than two views, (2) the computational cost, (3) missing evaluation details, and (4) the effect of view overlap. The rebuttal successfully addressed these concerns, convincing even the initially negative reviewer. Consequently, all reviewers recommended acceptance.